# Olive Oil (Royal Cultivar) from Mill Obtained by Short Time Malaxation and Early Ripening Stage

**DOI:** 10.3390/foods13162588

**Published:** 2024-08-18

**Authors:** Raúl Peralta, Francisco Espínola, Alfonso M. Vidal, Manuel Moya

**Affiliations:** 1Department Chemical, Environmental and Materials Engineering, Universidad de Jaén, Paraje Las Lagunillas, 23071 Jaén, Spain; rpgarcia@ujaen.es (R.P.); amvidal@ujaen.es (A.M.V.); mmoya@ujaen.es (M.M.); 2Center for Advanced Studies in Earth Sciences, Energy and Environment (CEACTEMA), Universidad de Jaén, 23071 Jaén, Spain

**Keywords:** extraction efficiency, virgin olive oil, aroma profiling, odor quality, SPME-GC, flavor, volatile compounds, (*E*)-2-hexenal, phenolic compounds, oleocanthal

## Abstract

The olive oil from the Royal cultivar has not been studied in depth, especially its relationship between analytical and sensory parameters. Currently, it is a minority cultivar, but due to its excellent organoleptic properties, it is constantly growing. The research objective is to obtain excellent-quality olive oil from the Royal cultivar at an industrial extraction plant and characterize the oil sensory and analytically. For this purpose, three important factors were set: very early olives; very low-time olive paste malaxation; and environmental temperature. The analytical parameters studied were volatile and phenolic compounds, fatty acids, photosynthetic pigments, and other quality parameters. Fourteen phenolic compounds were identified and found in significantly higher concentrations in Royal olive oil, including the oleacein compound. Moreover, volatile compounds from the LOX pathway, such as hexenal, (*E*)-2-hexenal, and (*Z*)-3-hexen-1-ol, had significantly higher concentrations, which were related to organoleptic characteristics: very fruity, not very spicy, and very low bitterness. The highest values obtained were 74.98% extraction efficiency at 30 min; 71.31 mg/kg chlorophyll content at 30 min; 156.38 mg/kg phenolic compound at 30 min; 18.98 mg/kg volatile compounds at 15 min; and better organoleptic characteristics at 15 min. The oil extraction efficiency was lower than that of other olive cultivars; nevertheless, the content of volatile compounds is higher.

## 1. Introduction

Spain is the world’s leading olive oil producer, and the Picual cultivar is the main one, although there are many different varieties of olives. One of them is the Royal cultivar, which can be located in some areas of Sierra de Cazorla (Jaén, Spain) [1]. This is a minority olive cultivar in Spain, but it is highly valued for its organoleptic properties. The olive oil obtained from Royal olives is an excellent virgin olive oil, very fruity, smooth, and pleasant to the palate. This is causing the growing area to constantly increase; however, few studies have been performed on the Royal olive oil. In an early study, Escuderos et al. [2] reported the composition, comparing it with eight different virgin olive oils from various producing countries, and recently Miho et al. [3] included Royal cultivars in a study of forty-four olive oils recognized by the International Olive Council (IOC) as some of the most widely distributed varieties globally.

Triglycerides are a major part of olive oil, although they have a small but no less important part of phenolic and volatile compounds, tocopherols, and pigments, among others. The antioxidant properties of phenolic compounds in olive oil contribute to important health benefits by protecting blood lipids from oxidative stress, according to the EEC [4]. Moreover, they have special interest because they affect oil aroma, taste, and stability [5]. Oleocanthal and oleacein have special relevance among the phenolic compounds. Said compounds appear during the production of olive oil from the hydrolysis of ligstroside and oleuropein, respectively [6]. In particular, oleocanthal is a compound found exclusively in olive oil and has anti-inflammatory properties similar to the action of ibuprofen. In addition, several potential bioactivity and health claims are attributed to this compound [7].

Additionally, the initial characteristics of olive oil noticed by consumers are its color and aroma, which are crucial attributes for the overall perception of food products. The color of olive oil is determined by pigments such as chlorophylls and carotenoids. Nevertheless, pigment composition can change significantly depending on the climate, ripening degree, and olive cultivars [8]. Volatile compounds are very interesting because they influence a lot of taste and aroma. During the production of olive oil, these compounds are mainly formed by the action of the enzymes released during the olive milling process. In addition, other factors influence the final content of volatile compounds that we find in olive oils, including climatic, agronomic, and technological factors [9,10,11]. Among them, two of the most important are malaxation time and temperature [12]. Other researchers have concluded that, in general, an increase in the malaxation temperature until around 30 °C is positive in reference to the phenol content of the oil [12,13,14]. While high malaxation time and low malaxation temperature increase the volatile compounds [15]. By controlling these factors, it is possible to produce high-quality olive oil that is rich in beneficial phenolic compounds and has a high content of volatile compounds. On the contrary, it has been developed as a novel procedure without the malaxation step through the use of non-thermal ultrasound-assisted extraction and pulsed electric field treatments, and its results in a continuous-flow industrial prototype showed interesting increased extraction yields [16].

The objective of our research was to obtain an excellent olive oil from the Royal cultivar from the consumer’s point of view, with special attention to volatile compounds, photosynthetic pigments, and phenolic compounds without compromising the yields. For this purpose, based on the bibliography and the experience of the millworkers with this kind of olive and maturity index, the following factors were set: very early olives, very low-time olive paste malaxation, and working environment temperature; and two malaxing times were used: 15 min and 30 min. The oils obtained were analyzed from the decanter outlet and the vertical centrifuge. In addition, the oils obtained will be compared with the bibliographic data of oils from other cultivars in Spain (Picual, Arbequina, Koroneiki, and Arbosana).

## 2. Materials and Methods

### 2.1. Olive Fruits and Olive Oil Mill Extraction

The Royal olives (*Olea europaea* L. cv. Royal) were picked up from “Sierra de Cazorla” (Jaen, Spain). The olives were mechanically harvested in mid-October 2023, coming from 10-year-old olive trees and organically grown with irrigation. Afterward, olive fruits with a very low maturity index were transported directly to the oil press. Table 1 shows the maturity indices and other olive characteristics. The method described by Espínola et al. [17] was used to determine the maturity index. The oil content of olive paste was determined using the Soxhlet method, and the moisture was determined after drying the olive paste at 105 °C for 24 h, following the procedure defined by the EEC [18]. The olive characteristics were considered homogeneous for the two tests that were performed. However, a small statistically significant difference was observed for the oil content.

Two batches of about 5900 kg of olive fruits were milled in two days, using a screen size of 6 mm. The olive mill operated at an average capacity of 3500 kg/h. A hammer crusher, a thermomalaxer with a vertical vessel of 4000 kg, a two-phase centrifugal decanter (Amenduni model Taurus due) operating at 3000 rpm, and a vertical centrifuge (Gea Westfalia model WSC20-01-5) operating at a maximum of 7500 rpm. No water or coadjuvant were used in the malaxer.

To calculate the extraction efficiency (%), percentage of oil extracted with respect to the total oil content present in the olive’s samples, pomace was collected at three different times during each experiment after crushing and from the decanter [19].

Trials were designed to produce high-quality virgin olive oil and to assess the efficacy of industrial extraction. Once the optimal values of the factors have been established according to the bibliography, olives with a low maturity index, the lowest possible malaxing temperature, in our case, room temperature, and the largest possible sieve size of the hammer crusher, 6 mm. In this research, two industrial tests were carried out at two different malaxing times as low as possible, 15 min and 30 min.

### 2.2. Olive Oil Quality Criteria and Photosynthetic Pigments

Quality criteria such as acidity, peroxides, and spectrophotometric indices (K232 and K270) were determined following the methods described in EEC [18] of the European Commission.

Pigments, chlorophylls, and carotenoids in oil samples were determined by the methodology of Minguez-Mosquera et al. [20]. Three grams of oil was dissolved in a flask with cyclohexane. A UV spectrophotometer, model Shimadzu UV-1800 (Kyoto, Japan), was used. Chlorophylls were measured using absorbance at 670 nm, and carotenoid pigments were measured using absorbance at 470 nm. The pigment content of the samples is expressed in milligrams of pigment per kilogram of oil.

### 2.3. Fatty Acids

The fatty acid content was analyzed according to the methodology of the EEC [18]. Vidal et al. [21] have described details of the analytical method. A sample of 0.1 g of olive oil was combined with a methanolic solution containing potassium hydroxide. A 7890B-GC gas chromatograph and HP-88 capillary column (Agilent Technologies, Santa Clara, CA, USA) were used. Helium was used as a carrier gas at a flow rate of 20 cm s^−1^, which is equivalent to 1.01 mL min^−1^. The injector temperature was 250 °C with a split ratio of 1:100. One microliter of sample was injected. The oven temperature was set at 100 °C for 5 min, then increased with a ramp of 4 °C min^−1^ to 240 °C and held at 240 °C for 30 min to completion. The temperature of the FID was 260 °C. A standard mixture of fatty acids from *Supelco* (Bellefonte, PA, USA) was used to identify the fatty acids in the oils.

### 2.4. Phenolic Compounds

Phenolic compounds in olive samples were identified and quantified using the method recommended by the International Olive Council [22]. The specifics of the analytical procedure are detailed in Vidal et al. [23]. The extraction of the phenolic components was performed directly from 2 g of olive oil using a methanolic solution (80%). A Shimadzu HPLC system (Shimadzu Corp., Kyoto, Japan) and a BDS Hypersil C18 column (Thermo Fisher Scientific, Waltham, MA, USA) were employed. The flow rate of the mobile phase was 1 mL min^−1^. A ternary gradient was made up of phosphoric acid–water to 0.2%, methanol, and acetonitrile. The column oven temperature was set to 30 °C. Twenty microliters of samples was injected. The wavelength for the UV detector was 280 nm. The results are reported as milligrams of tyrosol per kilogram of oil. Identification of the phenolic compounds was achieved by comparison with some analytical standards and the method of IOC [22].

### 2.5. Volatile Compounds

The method outlined by Vidal et al. [23] was employed, utilizing headspace (HS) and solid-phase microextraction (SPME) combined with gas chromatography (GC) and flame ionization detection (FID). The fiber of SPME is composed of carboxen/DVB/polydimethylsiloxane and measures 2 cm in length with a film thickness of 50/30 μm. It was acquired from Supelco (Bellefonte, PA, USA). A gas chromatograph, model 7890B, and a capillary column DB-WAXetr (Agilent Technologies, CA, USA) were used. The fiber adsorbed the volatiles from the 2 g sample of olive oil. The volatiles were desorbed into the injector at 260 °C. First, the oven was maintained at 40 °C for 10 min. Then, a ramp of 3 °C min^−1^ up to 160 °C, and another ramp of 15 °C min^−1^ up to 200 °C. The FID detector was 280 °C of temperature. The compound used as the internal standard was 4-methyl-2-pentanol, and another 39 external standards were used for the identification and quantification. The data are reported as milligrams of compound per kg of olive oil.

### 2.6. Sensorial Analysis

Sensorial analysis is the examination of organoleptic properties through the human senses. A panel of eight expert tasters conducted the analysis using the method recommended by the EEC [18], including Annex XII and the above amendments. The panelists are part of the Agrifood Laboratory (Granada, Spain), accredited under the EN ISO/IEC 17025:2017 standard [24]. The oils were evaluated to identify positive attributes such as fruitiness, pungency, and bitterness, as well as to detect any defects.

### 2.7. Statistical Analysis

Data were analyzed using StatGraphics Centurion XIX software (Statpoint Technologies, Inc., Warrenton, VA, USA). Fisher’s least significant differences (Fisher’s LSD) were calculated for each response variable. Means were considered significantly different at a *p*-value of less than 0.05.

## 3. Results and Discussion

### 3.1. Extraction Efficiency, Quality Criteria, and Pigment Composition

Table 2 indicates the values obtained for the extraction efficiency; this shows that the extraction efficiency increased with malaxing time, 74.98% at 30 min versus 73.13 at 15 min, which highlights the importance of this operation to increase the amount of oil obtained in the process of making it.

Regarding the parameters of quality, acidity, peroxide index, and spectrophotometric indices, although there are slight differences, no clear trend is observed; the acidity was reduced after vertical centrifugation for 30 min of malaxing but not for 15 min; the peroxide index increased after vertical centrifugation for 15 min of malaxing but not for 30 min; eventually, the K232 decreased for both malaxation times, but the K270 did not vary. In summary, the differences, although statistically significant in some cases, are very small and may be related to the health status of the olives.

The results for the contents of chlorophylls and carotenoids (Figure 1) show statistically significant differences. On the one hand, there are differences between the oils obtained from malaxing for 15 min and 30 min, and on the other hand, between the oils obtained at the decanter outlet and the vertical centrifuge outlet. In the first case, it is concluded that the content of photosynthetic pigments increases with malaxing time. Espínola et al. [25] and Vidal et al. [26] observed the same behavior working in an interval of 30 min to 90 min and different cultivars. In another study, Vidal et al. [23] obtained the same results working in an interval of 60 min to 120 min at the industrial level under continuous working conditions. In the second case, there is a difference between the oil coming from the vertical centrifuge and the decanter, possibly due to the fact that they are fat-soluble pigments that concentrate when the oils are clearer.

### 3.2. Fatty Acids

Table 3 presents the result of the determination of individual FA content. In addition, the sums of saturated fatty acids (SFA), polyunsaturated fatty acids (PUFA), and monounsaturated fatty acids (MUFA) were calculated. Likewise, the C18:1/C18:2 ratio and the MUFA/PUFA ratio were determined. Only statistically significant results for linoleic acid were found between 15 min and 30 min of malaxing.

The oleic acid content obtained from Royal oil, 74.92%, is slightly lower than that obtained by Escuderos et al. [2] for the same cultivar, 78.19%; however, the palmitic acid content, 14.28%, is higher. Benito et al. [27] also obtained a value of 79.70% for the same cultivar. With respect to other cultivars, the fatty acid composition is not a differentiating factor for Royal [21,25,28]. Furthermore, the values obtained were compared with the established ranges by the EU and IOC for virgin olive oils. The results obtained were aligned with the parameters established by both organizations.

### 3.3. Phenolic Compounds

Phenolic compounds are crucial in VOO due to their large antioxidant activity, which contributes to the typical bitter taste of the oils and especially to their shelf life [29]. For the phenolic profiling, ten compounds were evaluated, including the aglycones from ligstroside (p-HPEA-EA) and oleuropein (3,4-DHPEA-EA), along with their decarboxymethylated forms (p-HPEA-EDA and 3,4-DHPEA-EDA, respectively), the major phenolic compounds of the oils, as shown in Table 4. However, behind 3,4-DHPEA-EA and p-HPEA-EA there is a set of phenols, and we have not been able to discriminate between the closed monoaldehyde form of oleuropein aglycone and the open aldehyde form.

Table 4 presents the individual content of phenolic compounds determined by HPLC (mg/kg) and the total phenols determined by HPLC (mg/kg), which is the sum of the individual phenolic compounds. The total content of phenolic compounds, primarily secoiridoids, decreased by 11% after vertical centrifugation of the oil for 30 min of malaxing. An analysis of individual phenolic compounds showed that luteolin, hydroxytyrosol, and lignans did not suffer significant variations through centrifugation. On the other hand, tyrosol, vainillinic acid, ferulic acid, p-coumaric acid, and secoiridoids had a significant decrease during centrifugation. This reduction can be attributed to the migration of the hydrophilic phenols from the oil to the water and to the increased level of dissolved oxygen in the olive oil during centrifugation, which can lead to oxidation reactions in the phenolic compounds [30,31].

A clear increase is observed with the malaxation time, going from 96.99 mg/kg to 138.86 mg/kg when the time increases from 15 min to 30 min, although the variation is different for each phenolic compound; for example, a clear increase in oleacein is seen but not in oleocanthal. Similar results have been reported by other authors who studied malaxation on separately ground samples [23,32,33]. However, Ben Brahim et al. [34] found that malaxation time did not significantly affect phenolic content.

Comparing our results with those obtained by Miho et al. [3] for the same cultivar, “Royal de Cazorla”, our values of the total content of phenolic compounds are lower, 156 mg/kg compared to 290 mg/kg; however, we must take into account that they worked at a laboratory scale and their value is an average of three crop seasons. On the other hand, Miho et al. [3] report 44 monovarietal phenolic profiles of virgin olive oils, and Royal has the lowest value, so it is expected that the oil produced by the Royal cultivar will be very sweet, slightly bitter, and pungent.

### 3.4. Volatile Compounds

Volatile compounds are a crucial group of molecules responsible for aroma. Smell is a key factor in the acceptance or rejection of various foods. Volatile compounds are detected through the nose and recall numerous associations and emotions.

The wide variety of volatile compounds responsible for the fruity and green aroma in olive oil are produced [35], essentially through the so-called lipoxygenase (LOX) pathway. They generally have a low molecular mass and high volatility, are sparingly soluble in water but soluble in oil and ethanol, and have certain chemical characteristics to bind with specific proteins [36]. The lipoxygenase pathway uses linoleic and linolenic fatty acids as substrates [37] and the enzyme lipoxygenase (LOX), present in olive pulp. An attempt has been made to relate the volatile compounds present in virgin olive oil with the sensory attributes that people perceive. A comprehensive description of the role of a large number of these compounds in the aroma of oil can be obtained from the literature reviews [38,39,40].

The main volatile compounds detected and quantified in the oils obtained are shown in Table 5. Nine from the route lipoxygenase (LOX) pathway; two aldehydes (hexanal and (*E*)-2-hexenal); five alcohols (hexan-1-ol, (*E*)-2-hexen-1-ol, (*Z*)-3-hexen-1-ol, 1-penten-3-ol, and (*Z*)-2-penten-1-ol). One ketone (1-penten-3-one), and one ester ((*Z*)-3-hexenyl acetate). The main volatile compounds detected were (*E*)-2-hexenal, ranging from 4.63 to 5.12 mg/kg, followed by (*Z*)-3-hexen-1-ol.

There are no statistically significant differences in the total content of volatile compounds after vertical centrifugation of the oils, nor between the oils obtained at 15 min or 30 min of malaxation. However, at the individual level, there was a slight decrease in hexanal, hexan-1-ol, and (*Z*)-3-hexenyl acetate, probably offset by the slight increase in (*E*)-2-hexen-1-ol and 1-penten-3-one, but without statistically significant differences. Most authors have found that an increase in mixing time increases the concentration of volatile compounds [32,41,42,43]. Youssef et al. [44], however, in some cases observed an opposite behavior, for example, for the oil of Morisca [45].

Studies conducted by Gómez-Rico et al. [46] regarding the profiles of volatile compounds, obtained from various varieties of Spanish olives, showed that they have a similar qualitative composition, although very different from a quantitative point of view. Only the Arbequina cultivar was superior to the Royal cultivar.

In a bibliographic review carried out by Aparicio et al. [47], of 35 monovarietal oils from 7 countries in the Mediterranean basin, only Coratina and Cima di Bitonto from Italy and Lechín from Spain showed a higher concentration of volatile compounds, so it can be concluded that the oil of the Royal variety is among the best oils in terms of content of volatile compounds.

### 3.5. Sensory Analysis

The sensory analysis of the oils is shown in Table 6. The sensory profile shows an increase in the fruity attribute of the oils from the outlet of the vertical centrifuge with respect to the oils from the outlet of the decanter, but it is not observed in the bitter and spicy attributes. On the other hand, statistically significant differences are only observed in the bitter attribute between the oils obtained after malaxation of the pasta for 15 min and 30 min, resulting in the second case being more bitter. However, from the sensory profiles, it can be seen that the olive oil of the Royal variety is very balanced, that is, fruity and not very spicy or bitter. Benito et al. [27] obtained similar results with the same cultivar.

The perceived aroma cannot be attributed to a single compound but rather to a mixture of several volatile compounds. However, not all compounds contribute equally to the aroma of the oil. The impact of each compound depends on its odor threshold, among other factors [48]. Therefore, the odor activity value (OAV), which is the concentration of a volatile compound divided by its odor threshold, allows for an estimate of each compound´s contribution to the overall aroma of the oil [49,50]. However, using each OAV to tentatively determine the aromatic profile of an oil results in a complex model that is difficult to interpret. However, volatile compounds with similar odor descriptors can be grouped according to the criteria described by Reboredo-Rodríguez et al. [28]. In our case, we have made three groups: fruity, bitter, and spicy. The total OAV of each group was calculated by adding the OAV calculated for each volatile compound belonging to a particular group. This approach can be useful to link the quantitative data from the chemical analysis of the volatiles with the sensory experience to form a preliminary aroma profile. It is important to recognize that certain volatile compounds, such as 1-penten-3-one, may contribute to the bitter and pungent sensations, although these sensory attributes are generally attributed to the stimulation of taste cells and the trigeminal nerve by phenolic compounds rather than volatile compounds such as secoiridoids [51].

Figure 2 shows the contribution of each attribute to the sensory profile of the oils obtained, together with the results of the test panel. Considering the results, there is a good relationship between sensory and analytical analysis (OAV values). In reality, the intensities of the perceived sensory attributes (panel test results) were in substantial agreement with the analytical data (OAV), except for the bitterness attribute.

The spicy sensation perceived by tasters is mainly caused by phenolic compounds (secoiridoids) and not by the volatile substances grouped in the “spicy” category. This sensation can be reinforced by volatiles grouped as “spicy” but cannot be predictive on its own. Moreover, the acetic acid present in the oils could alter some of these sensations. Since only C5 and C6 compounds were used to construct the odorant series, this compound was not considered [52]. Therefore, both sensory and analytical data are more appropriate.

## 4. Conclusions

An evaluation of the composition and sensory analysis of the oils obtained from the Royal variety has been conducted. Their compositions of fatty acids match with respect to other olive cultivars. The oils generally exhibited a medium content of total phenols and pigments. Due to this, these oils are likely to be highly stable, particularly in dark storage conditions. Moreover, chlorophylls reinforced the oil’s stability. Regarding volatile compounds, the most prevalent in all samples were those from the C6 aldehyde fraction, which is characteristic of high-quality olive oils. Fruity stood out as the main sensory attribute in the oils, followed by pungency and bitterness. The effect of the time of malaxation of the paste and the vertical centrifugation of the oil was evaluated on the cultivar Royal. The quality parameters such as acidity, peroxides, and ultraviolet ray absorbance were not altered by the mixing time, but the extraction efficiency and the chlorophyll content increased in continuous industrial production. Volatile compounds were affected by changes in whipping time. Total phenol content and several phenolic compounds, such as oleacein and oleocanthal, increased with malaxation time. Regarding the sensory analysis described, the results did not show notable differences in terms of fruitiness and pungency, although higher bitterness values were observed for longer malaxing times. However, it is concluded that the olive oil of the Royal variety is smooth, aromatic, and very balanced.

## Figures and Tables

**Figure 1 foods-13-02588-f001:**
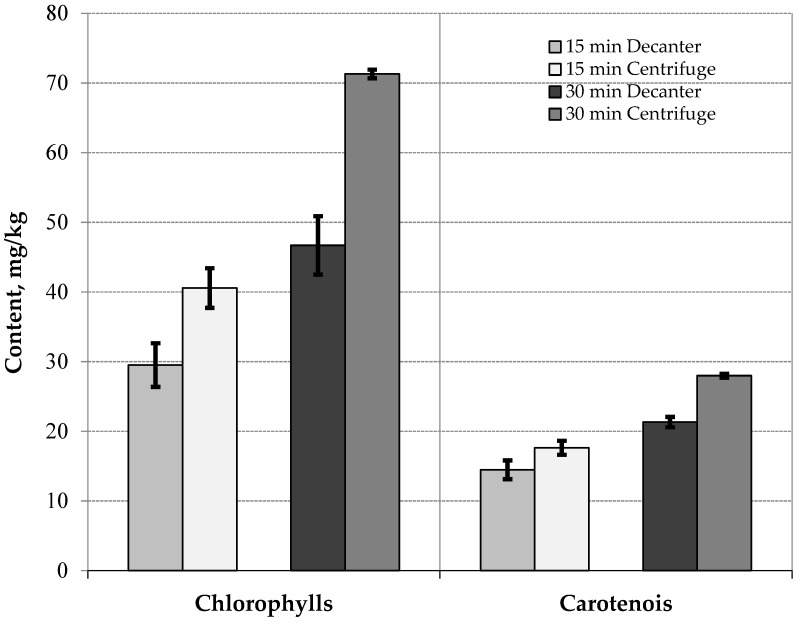
Content of chlorophylls and carotenoids (mg/kg) of oils obtained at the decanter outlet and the vertical centrifuge outlet with different malaxing times of 15 min and 30 min.

**Figure 2 foods-13-02588-f002:**
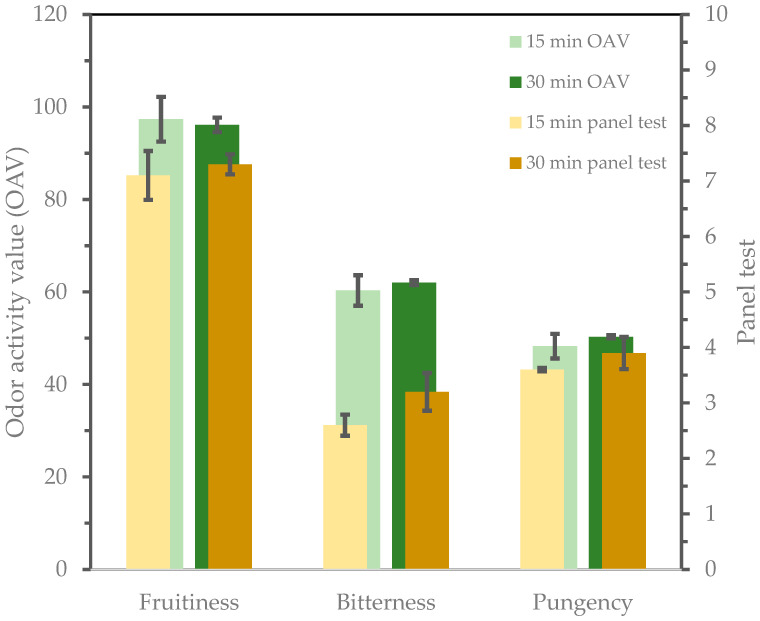
Contribution of each attribute to the sensory profile of oils obtained with different malaxing times of 15 min and 30 min. Comparison of contribution measured by the addition of the odor activity value (OAV) of the volatile compounds that belongs to each attribute group and the sensory attributes perceived from the panel test.

**Table 1 foods-13-02588-t001:** The parameters of the industrial trials and the characteristics of the olive fruit.

			Olive Characteristics *			
Trial	Time, min	Temperature, °C	Maturity Index	Oil, %	Moisture, %	Solids, %
1	15	32.6 ± 1.3	1.07 ± 0.01	14.33 ± 0.07 ^a^	60.06 ± 0.35	25.61 ± 0.42
2	30	31.6 ± 1.2	1.05 ± 0.02	13.52 ± 0.12 ^b^	60.52 ± 0.50	25.96 ± 0.40

* Values are presented as mean ± SD. Different letters ^(a, b)^ denoting Fisher’s least significant differences (LSD) (*p*-value = 0.0006, Fisher’s LSD = 0.23). Statistically significant differences are noted at the 95% of confidence level.

**Table 2 foods-13-02588-t002:** Oil extraction efficiency and quality criteria for oil samples before (decanter exit) and after vertical centrifugation (centrifuge exit) *.

	Trial 1 (15 min)		Trial 2 (30 min)		*p*-Value	Fisher’s LSD
Extraction efficiency (%)	73.13 ± 0.28 ^a^		74.98 ± 0.31 ^b^		0.0016	0.67
Oils	Decanter	Centrifuge	Decanter	Centrifuge		
Acidity (%)	0.31 ± 0.01 ^a^	0.30 ± 0.01 ^a^	0.28 ± 0.01 ^b^	0.24 ± 0.01 ^c^	0.0001	0.02
Peroxides (mEq O_2_/kg)	3.71 ± 0.20 ^a^	4.44 ± 0.24 ^b^	2.96 ± 0.12 ^c^	3.10 ± 0.14 ^c^	0.0000	0.35
K232	1.70 ± 0.07 ^a^	1.57 ± 0.07 ^b^	1.73 ± 0.02 ^a^	1.58 ± 0.03 ^b^	0.0113	0.10
K270	0.17 ± 0.02 ^a^	0.16 ± 0.03 ^a^	0.15 ± 0.02 ^a^	0.15 ± 0.02 ^a^	0.8711	
Chlorophylls (mg/kg)	29.51 ± 3.13 ^a^	40.57 ± 2.85 ^b^	46.70 ± 4.19 ^c^	71.31 ± 0.62 ^d^	0.0000	5.63
Carotenoids (mg/kg)	14.47 ± 1.35 ^a^	17.63 ± 1.01 ^b^	21.32 ± 0.74 ^c^	27.98 ± 0.27 ^d^	0.0000	1.76

* Values are presented as mean ± SD. Different letters ^(a, b, c, d)^ denoting Fisher’s least significant differences (LSD). Statistically significant differences are noted at the 95% of confidence level.

**Table 3 foods-13-02588-t003:** Fatty acid content of Royal olive oils before and after vertical centrifugation. Data are expressed in % *w*/*w* methyl ester *.

	Trial 1 (15 min)		Trial 2 (30 min)		*p*-Value	Fisher’s LSD
	Decanter	Centrifuge	Decanter	Centrifuge		
Palmitic acid C16:0	14.40 ± 0.19 ^a^	14.47 ± 0.03 ^a^	14.16 ± 0.02 ^b^	14.28 ± 0.16 ^a,b^	0.0667	0.24
Palmitoleic acid C16:1	1.27 ± 0.03 ^a,b^	1.29 ± 0.00 ^b^	1.23 ± 0.00 ^a^	1.25 ± 0.03 ^a,b^	0.0833	0.04
Heptadecanoic acid C17:0	0.04 ± 0.00	0.04 ± 0.00	0.04 ± 0.00	0.04 ± 0.00		
Heptadecenoic acid C17:1	0.09 ± 0.00	0.09 ± 0.00	0.09 ± 0.00	0.09 ± 0.00		
Stearic acid C18:0	1.41 ± 0.02 ^a,b^	1.43 ± 0.00 ^b^	1.39 ± 0.00 ^a^	1.40 ± 0.02 ^a,b^	0.0598	0.03
Oleic acid C18:1	73.97 ± 1.68 ^a,b^	73.20 ± 0.21 ^a^	75.77 ± 0.06 ^b^	74.92 ± 1.18 ^a,b^	0.0690	1.95
Linoleic acid C18:2	8.88 ± 0.06 ^a^	8.72 ± 0.19 ^a^	6.56 ± 0.05 ^b^	6.68 ± 0.03 ^b^	0.0000	0.26
Linolenic acid C18:3	0.26 ± 0.00	0.26 ± 0.00	0.26 ± 0.00	0.26 ± 0.00		
Arachidic acid C20:0	0.04 ± 0.02	0.04 ± 0.02	0.05 ± 0.00	0.05 ± 0.01		
Eicosenoic acid C20:1	0.32 ± 0.00	0.32 ± 0.00	0.32 ± 0.00	0.32 ± 0.00		
Behenic acid C22:0	0.08 ± 0.00	0.08 ± 0.00	0.08 ± 0.00	0.08 ± 0.00		
Lignoceric acid C24:0	0.06 ± 0.00	0.06 ± 0.00	0.06 ± 0.00	0.06 ± 0.00		
SFA	16.03 ± 0.21 ^a^	16.12 ± 0.03 ^a^	15.77 ± 0.02 ^b^	15.91 ± 0.17 ^a,b^	0.0632	0.26
MUFA	75.65 ± 1.65 ^a,b^	74.90 ± 0.21 ^a^	77.42 ± 0.06 ^b^	76.59 ± 1.15 ^a,b^	0.0688	1.90
PUFA	9.15 ± 0.04 ^a^	9.09 ± 0.19 ^a^	6.82 ± 0.05 ^b^	6.94 ± 0.02 ^b^	0.0000	0.33
C18:1/C18:2	8.33 ± 0.19 ^a^	8.40 ± 0.21 ^a^	11.56 ± 0.10 ^b^	11.22 ± 0.18 ^c^	0.0000	0.33
MUFA/PUFA	8.27 ± 0.19 ^a^	8.34 ± 0.20 ^a^	11.36 ± 0.10 ^b^	11.03 ± 0.17 ^c^	0.0000	0.32

* Values are presented as mean ± SD. Different letters ^(a, b, c)^ denoting Fisher’s least significant differences (LSD). Statistically significant differences are noted at the 95% of confidence level. Monounsaturated fatty acid (MUFA); polyunsaturated fatty acid (PUFA); saturated fatty acid (SFA).

**Table 4 foods-13-02588-t004:** Before and after vertical centrifugation of individual phenolic compounds, expressed in mg tyrosol/kg oil *.

	Trial 1 (15 min)		Trial 2 (30 min)		*p*-Value	Fisher’s LSD
	Decanter	Centrifuge	Decanter	Centrifuge		
Phenolic alcohols						
Hydroxytyrosol	1.34 ± 0.57 ^a^	1.16 ± 0.18 ^a,b^	0.65 ± 0.14 ^a,b^	0.55 ± 0.17 ^b^	0.0963	0.75
Tyrosol	2.08 ± 0.17 ^a^	1.25 ± 0.18 ^b^	2.39 ± 0.15 ^c^	1.45 ± 0.06 ^b^	0.0002	0.34
Phenolic acids						
Vainillinc acid	0.86 ± 0.09 ^a^	0.48 ± 0.09 ^b^	0.81 ± 0.06 ^a^	0.53 ± 0.09 ^b^	0.0019	0.15
Vainillin	1.80 ± 0.05 ^a^	1.31 ± 0.14 ^b^	2.07 ± 0.15 ^c^	--	0.0000	0.24
*p*-Coumaric acid	1.51 ± 0.36 ^a^	2.44 ± 1.22 ^a,b^	6.94 ± 0.02 ^c^	4.20 ± 0.76 ^b^	0.0067	2.06
Ferulic acid	6.78 ± 0.11 ^a^	1.87 ± 0.14 ^b^	8.40 ± 0.37 ^c^	5.79 ± 0.46 ^d^	0.0000	0.83
Secoiridoids						
3.4-DHPEA-EDA (oleacein)	20.48 ± 2.88 ^a^	25.61 ± 1.05 ^a^	53.33 ± 1.33 ^b^	48.42 ± 7.01 ^b^	0.0004	8.22
3.4-DHPEA-EA	17.80 ± 0.68 ^a^	16.37 ± 0.47 ^a^	21.38 ± 2.02 ^b^	21.06 ± 0.13 ^b^	0.0079	2.75
p-HPEA-EDA (oleocanthal)	18.45 ± 0.92 ^a^	17.08 ± 2.81 ^a^	28.71 ± 1.64 ^b^	24.48 ± 3.18 ^b^	0.0033	5.17
p-HPEA-EA	17.96 ± 2.25 ^a^	17.37 ± 2.40 ^a^	16.63 ± 0.59 ^a^	15.80 ± 2.51 ^a^	0.6787	
Lignans						
Pinoresinol + Acetoxypinoresinol	8.85 ± 0.72 ^a,b^	6.93 ± 0.65 ^a^	7.83 ± 0.83 ^a,b^	9.08 ± 0.90 ^b^	0.0910	1.80
Flavones						
Luteolin	1.13 ± 0.56 ^a^	0.56 ± 0.23 ^a^	1.22 ± 0.32 ^a^	1.77 ± 0.85 ^a^	0.1341	
Apigenin	8.01 ± 0.65 ^a^	4.40 ± 0.07 ^b^	5.45 ± 0.37 ^b^	7.46 ± 0.41 ^a^	0.0009	1.09
Total HPLC Phenols	105.54 ± 2.85 ^a^	96.99 ± 6.18 ^b^	156.38 ± 1.12 ^c^	138.86 ± 4.60 ^d^	0.0000	7.80

* Values are presented as mean ± SD. Different letters ^(a, b, c, d)^ denoting Fisher’s least significant differences (LSD). Statistically significant differences are noted at the 95% of confidence level.

**Table 5 foods-13-02588-t005:** Individual content of volatile compounds before and after vertical centrifugation. Expressed in mg/kg oil *.

	Trial 1 (15 min)		Trial 2 (30 min)		*p*-Value	Fisher’s LSD
	Decanter	Centrifuge	Decanter	Centrifuge		
LOX pathway						
Hexanal	2.00 ± 0.18 ^a^	1.85 ± 0.04 ^a,b^	1.75 ± 0.09 ^b^	1.69 ± 0.08 ^b^	0.0397	0.21
Hexan-1-ol	1.28 ± 0.05 ^a^	1.31 ± 0.06 ^a^	1.03 ± 0.05 ^b^	1.08 ± 0.07 ^b^	0.0008	0.11
(*E*)-2-Hexenal	5.03 ± 0.24	5.12 ± 0.40	4.63 ± 0.29	4.94 ± 0.09	0.3357	
(*E*)-2-Hexen-1-ol	1.36 ± 0.11	1.32 ± 0.12	1.40 ± 0.03	1.37 ± 0.08	0.7338	
(*Z*)-3-Hexen-1-ol	2.85 ± 0.09 ^a^	2.97 ± 0.16 ^a^	2.50 ± 0.24 ^b^	2.94 ± 0.12 ^a^	0.0285	0.31
(*Z*)-3-Hexenyl acetate	1.69 ± 0.07 ^a^	1.90 ± 0.16 ^b^	1.48 ± 0.08 ^c^	1.44 ± 0.04 ^c^	0.0044	0.22
1-Penten-3-ol	1.16 ± 0.20	1.08 ± 0.18	1.09 ± 0.13	1.11 ± 0.05	0.9190	
1-Penten-3-one	2.27 ± 0.09 ^a,b^	2.28 ± 0.11 ^a,b^	2.03 ± 0.24 ^a^	2.38 ± 0.01 ^b^	0.0793	0.27
(*Z*)-2-Penten-1-ol	1.08 ± 0.07 ^a,b^	1.15 ± 0.05 ^b^	0.97 ± 0.12 ^a^	1.22 ± 0.03 ^b^	0.0234	0.15
Total LOX, mg/kg	18.72 ± 0.64 ^a^	18.98 ± 0.58 ^a^	16.90 ± 0.88 ^b^	18.16 ± 0.47 ^a^	0.0197	1.24
Other compounds						
Acetic acid	0.65 ± 0.07 ^a^	1.03 ± 0.14 ^a,b^	1.00 ± 0.19 ^a,b^	0.96 ± 0.03 ^b^	0.1379	0.32
(*E*)-2-Pentenal	0.78 ± 0.05 ^a,b^	0.80 ± 0.04 ^a,b^	0.73 ± 0.09 ^a^	0.85 ± 0.04 ^b^	0.1919	0.11
Pentan-3-one	0.47 ± 0.04	0.47 ± 0.04	0.46 ± 0.06	0.53 ± 0.02	0.3032	
Octanal	0.75 ± 0.08	0.75 ± 0.07	0.73 ± 0.07	0.78 ± 0.07	0.9009	

* Values are presented as mean ± SD. Different letters ^(a, b, c)^ indicate Fisher’s least significant differences (LSD). Statistically significant differences are noted at the 95% of confidence level.

**Table 6 foods-13-02588-t006:** Sensory characteristics for the oil samples before and after vertical centrifugation (medians) *.

	Trial 1 (15 min)		Trial 2 (30 min)		*p*-Value	Fisher’s LSD
	Decanter	Centrifuge	Decanter	Centrifuge		
Fruitiness	6.0 ± 0.43 ^a^	7.1 ± 0.44 ^b^	6.5 ± 0.72 ^a,b^	7.3 ± 0.18 ^b^	0.0393	0.91
Bitterness	2.7 ± 0.27 ^a,b^	2.6 ± 0.19 ^a^	3.4 ± 0.32 ^c^	3.2 ± 0.34 ^b,c^	0.0243	0.54
Pungency	3.5 ± 0.20 ^a^	3.6 ± 0.03 ^a^	3.9 ± 0.24 ^a^	3.9 ± 0.29 ^a^	0.1091	0.40

* Values are presented as median ± asymptotic robust standard deviation. Different letters ^(a, b, c)^ denoting Fisher’s least significant differences (LSD). Statistically significant differences are noted at the 95% of confidence level.

## Data Availability

The original contributions presented in the study are included in the article, further inquiries can be directed to the corresponding author.

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
