# Peer review of "Olive Oil (Royal Cultivar) from Mill Obtained by Short Time Malaxation and Early Ripening Stage"

_foods, 2024, doi:10.3390/foods13162588_

Round 1
Reviewer 1 Report
Comments and Suggestions for Authors
This study reports on the effect of various factors on the quality of olive oil obtained from Royal varieties. The factors studied included very early ripening olives, very short olive paste relaxation time and ambient temperature. The parameters analysed were mainly volatile and phenolic compounds, fatty acids, photosynthetic pigments and other quality parameters. Finally, the optimal processing conditions were determined. The manuscript is thematically clear, well thought out and well written. The following are some comments on the manuscript.
1. Similar research reports from the point of view of oil processing processes, methods, etc. should be added to the introduction section to enhance the quality of the preface section of the manuscript.
2. The specificity of the experimental samples led the authors to set the values mentioned in the manuscript for the selection of process parameters. The basis for the selection of these conditions should then be added to the text. In addition, it is necessary to expand the above parameters for the subsequent actual process to further meet the needs of production.
3. The influence of the different parameters studied on the quality of olive oil should be added to the text. In addition, the trend of the influence of the above indicators with different processes should be analysed in the Results and Discussion section in order to guide the optimisation of the process.
Comments on the Quality of English LanguageMinor editing of English language required.
Author Response
This study reports on the effect of various factors on the quality of olive oil obtained from Royal varieties. The factors studied included very early ripening olives, very short olive paste relaxation time and ambient temperature. The parameters analysed were mainly volatile and phenolic compounds, fatty acids, photosynthetic pigments and other quality parameters. Finally, the optimal processing conditions were determined. The manuscript is thematically clear, well thought out and well written. The following are some comments on the manuscript.
Thank you very much for your review, your comments have helped us to improve the manuscript. The authors have made various changes within the text.
Comment 1. Similar research reports from the point of view of oil processing processes, methods, etc. should be added to the introduction section to enhance the quality of the preface section of the manuscript.
Response 1. Thank you, in the introduction we have added a paragraph, from line 60 to 65, with several citations to the work of other researchers.
Comment 2. The specificity of the experimental samples led the authors to set the values mentioned in the manuscript for the selection of process parameters. The basis for the selection of these conditions should then be added to the text. In addition, it is necessary to expand the above parameters for the subsequent actual process to further meet the needs of production.
Response 2. Thank you, in lines 73 and 74 we indicate that the selected parameters are based on bibliographic data and the experience of the millworkers with this kind of olive and maturity index.
Comment 3. The influence of the different parameters studied on the quality of olive oil should be added to the text. In addition, the trend of the influence of the above indicators with different processes should be analysed in the Results and Discussion section in order to guide the optimisation of the process.
Response 3. Thank you, we have added your suggestions in the section discussion of results in the suggested sense, lines 233 to 236, 244 to 246 and 314 to 325.
Comments on the Quality of English Language
Minor editing of English language required.
Thanks for your feedback, there have been several changes in this regard.
Reviewer 2 Report
Comments and Suggestions for Authors
The manuscript "Olive oil (Royal cultivar) from mill obtained by short time malaxation and early ripening stage" presents research on the quality of royal olive oil depending on the processing time. The authors used samples from industrial production for their research, but did not specify the frequency of sampling the research material. The extraction process took two days. Overall, the research was carried out correctly, but the manuscript requires some details. A set of notes has been attached to the file.

Author Response
The manuscript "Olive oil (Royal cultivar) from mill obtained by short time malaxation and early ripening stage" presents research on the quality of royal olive oil depending on the processing time. The authors used samples from industrial production for their research, but did not specify the frequency of sampling the research material. The extraction process took two days. Overall, the research was carried out correctly, but the manuscript requires some details. A set of notes has been attached to the file.
Thank you very much for your review, your comments have helped us to improve the manuscript. The authors have responded to your individual comments and have made various changes within the text.
Comment 1. What size were the research samples?
Response 1. It is described in line 96. "Two batches of about 5900 kg of olive fruits were milled in two days"
Comment 2. How long did it take to dry and using what method?
Response 2. It takes 24 hours; this information has been added in the manuscript.
Comment 3. Describe the method briefly
Response 3. The methodology has been extended in the manuscript.
Comment 4. Describe the method briefly and the operating parameters of the chromatograph.
Response 4. The methodology has been extended in the manuscript.
Comment 5. Describe the method briefly and the operating parameters of the chromatograph. If phenolic compounds were identified on the basis of standards, why were their contents not given in mg? Secondly, how was it converted into tyrosol, since it is one of the phenolic acids?
Response 5. The methodology has been extended in the manuscript.
Phenolic compounds in olive samples were identified and quantified using the method recommended by the International Olive Council [22], and the results are reported as milligram of tyrosol per kilogram of oil. Only the identification of some phenolic compounds was performed with some analytical standards to reinforce the COI method [22], but the quantification was according to the method.
Comment 6. Describe the method briefly and the operating parameters of the chromatograph.
Response 6. The methodology has been extended in the manuscript.
Comment 7. To use Fisher's post-hoc test, you must first perform an analysis of variance, which tells you whether the post-hoc test can be used.
Response 7. Thank you very much for your comment. Effectively Fisher's test should be performed after an analysis of variance.
The software used, Statgraphics, allows comparison of data sets using the "simple ANOVA" menu. This allows the simultaneous performance of different statistical analyses, such as analysis of variance, multiple range test, Kruskal-Wallis test, etc., and the construction of different graphs, such as the means graph. The Statgraphics multiple range test and means graph "is based on Fisher's Least Significant Difference (LSD) procedure".
In the analysis of the experimental data, before using any statistical parameter, we have always observed that there is a statistically significant difference between the means, using analysis of variance. Although, sometimes, the p-value may be greater than 0.05, which would indicate that there are no statistically significant differences, but the multiple range test gave differences between the means. See Tables 3, 4 and 5.
The manuscript has been modified to include two columns in the different tables, one for p-value and one for Fisher's LSD. In cases where it is clear that the means are equal, no statistical parameter has been included, in others only the p-value has been included because there are no significant differences between the means, and in the others both parameters and letters, as superscripts, have been included. Equal letters indicate that there are no significant differences, different letters indicate that there are statistically significant differences between the means.
To include the two statistical columns greatly improves the quality of the manuscript, thanks again for your comment.
Round 2
Reviewer 1 Report
Comments and Suggestions for Authors
Adequate revisions have been made in the current manuscript.
Comments on the Quality of English LanguageEnglish language fine. No issues detected
Author Response
Thank you very much for your comments, the manuscript has improved considerably.